# Human Emotion Recognition: Review of Sensors and Methods

**DOI:** 10.3390/s20030592

**Published:** 2020-01-21

**Authors:** Andrius Dzedzickis, Artūras Kaklauskas, Vytautas Bucinskas

**Affiliations:** 1Faculty of Mechanics, Vilnius Gediminas Technical University, J. Basanaviciaus g. 28, LT-03224 Vilnius, Lithuania; andrius.dzedzickis@vgtu.lt; 2Faculty of Civil engineering, Vilnius Gediminas Technical University, Sauletekio ave. 11, LT-10223 Vilnius, Lithuania; arturas.kaklauskas@vgtu.lt

**Keywords:** human emotions, emotion perception, physiologic sensors

## Abstract

Automated emotion recognition (AEE) is an important issue in various fields of activities which use human emotional reactions as a signal for marketing, technical equipment, or human–robot interaction. This paper analyzes scientific research and technical papers for sensor use analysis, among various methods implemented or researched. This paper covers a few classes of sensors, using contactless methods as well as contact and skin-penetrating electrodes for human emotion detection and the measurement of their intensity. The results of the analysis performed in this paper present applicable methods for each type of emotion and their intensity and propose their classification. The classification of emotion sensors is presented to reveal area of application and expected outcomes from each method, as well as their limitations. This paper should be relevant for researchers using human emotion evaluation and analysis, when there is a need to choose a proper method for their purposes or to find alternative decisions. Based on the analyzed human emotion recognition sensors and methods, we developed some practical applications for humanizing the Internet of Things (IoT) and affective computing systems.

## 1. Introduction

With the rapid increase in the use of smart technologies in society and the development of the industry, the need for technologies capable to assess the needs of a potential customer and choose the most appropriate solution for them is increasing dramatically. Automated emotion evaluation (AEE) is particularly important in areas such as: robotics [1], marketing [2], education [3], and the entertainment industry [4]. The application of AEE is used to achieve various goals:(i)in robotics: to design smart collaborative or service robots which can interact with humans [5,6,7];(ii)in marketing: to create specialized adverts, based on the emotional state of the potential customer [8,9,10];(iii)in education: used for improving learning processes, knowledge transfer, and perception methodologies [11,12,13];(iv)in entertainment industries: to propose the most appropriate entertainment for the target audience [14,15,16,17].

In the scientific literature are presented numerous attempts to classify the emotions and set boundaries between emotions, affect, and mood [18,19,20,21]. From the prospective of automated emotion recognition and evaluation, the most convenient classification is presented in [3,22]. According to the latter classification, main terms defined as follows:(i)“emotion” is a response of the organism to a particular stimulus (person, situation or event). Usually it is an intense, short duration experience and the person is typically well aware of it;(ii)“affect” is a result of the effect caused by emotion and includes their dynamic interaction;(iii)“feeling” is always experienced in relation to a particular object of which the person is aware; its duration depends on the length of time that the representation of the object remains active in the person’s mind;(iv)“mood” tends to be subtler, longer lasting, less intensive, more in the background, but it can affect affective state of a person to positive or negative direction.

According to the research performed by Feidakis, Daradoumis and Cabella [21] where the classification of emotions based on fundamental models is presented, exist 66 emotions which can be divided into two groups: ten basic emotions (anger, anticipation, distrust, fear, happiness, joy, love, sadness, surprise, trust) and 56 secondary emotions. To evaluate such a huge amount of emotions, it is extremely difficult, especially if automated recognition and evaluation is required. Moreover, similar emotions can have overlapping parameters, which are measured. To handle this issue, the majority of studies of emotion evaluation focuses on other classifications [3,21], which include dimensions of emotions, in most cases valence (activation—negative/positive) and arousal (high/low) [23,24], and analyses only basic emotions which can be defined more easily. A majority of researches use variations of Russel’s circumplex model of emotions (Figure 1) which provides a distribution of basic emotions in two-dimensional space in respect of valence and arousal. Such an approach allows for the definition of a desired emotion and evaluating its intensity just analyzing two dimensions.

Using the above-described model, the classification and evaluation of emotions becomes clear, but still there are many issues related to the assessment of emotions, especially the selection of measurement and results evaluation methods, the selection of measurement hardware and software. Moreover, the issue of emotion recognition and evaluation remains complicated by its interdisciplinary nature: emotion recognition and strength evaluation are the object of psychology sciences, while the measurement and evaluation of human body parameters are related with medical sciences and measurement engineering, and sensor data analysis and solution is the object of mechatronics. 

This review focuses on the hardware and methods used for automated emotion recognition, which are applicable for machine learning procedures using obtained experimental data analysis and automated solutions based on the results of these analyses. This study also analyzes the idea of humanizing the Internet of Things and affective computing systems, which has been validated by systems developed by the authors of this research [25,26,27,28].

Intelligent machines with empathy for humans are sure to make the world a better place. The IoT field is definitely progressing on human emotion understanding thanks to achievements in human emotion recognition (sensors and methods), computer vision, speech recognition, deep learning, and related technologies [29].

## 2. Emotions Evaluation Methods

Emotion evaluations methods which are presented in the literature can be classified into two main groups according to the basic techniques used for emotions recognition: self-repot techniques based on emotions self-assessment by filing various questionnaires [30,31,32]; machine assessment techniques based on measurements of various parameters of human body [33,34,35]. In addition, there are frequent cases of simultaneous use of several methods in order to increase reliability of obtained results. According to [36,37], each emotion can be evaluated by analyzing five main components of emotion (Behavioral tendencies, physiological reactions, motor expressions cognitive appraisals and subjective feelings) but only the first four can be evaluated automatically and can give indications about the emotional state of an user during an interaction, without interrupting it. Subjective feelings usually evaluated only using self-assessment techniques.

Automated emotion recognition is typically performed by measuring various human body parameters or electric impulses in the nervous system and analyzing their changes. The most popular techniques are electroencephalography, skin resistance measurements, blood pressure, heart rate, eye activity, and motion analysis. 

### 2.1. Electroencephalography (EEG)

The EEG is an electrophysiological noninvasive technique for the recording of electrical activity arising from the human brain [38]. The first report on the application of this technique presented by Hans Berger, a German psychiatrist, pioneered the EEG in humans in 1924 [38]. EEG signals usually are collected using a special device called an electroencephalogram. The main parts of this device are special metal plate electrodes which should be placed on the human scalp, while in special cases, alternative needle electrodes can be inserted directly into the scalp [39]. In most cases, 8, 16 or 32 pairs of electrodes are located on four standard positions on the head: the nasion, inion, and right and left preauricular points (Figure 2a) [39] 

Electrodes can be attached to the human scalp using adhesive-conducting gel or special headsets [41] (Figure 2b) with installed electrodes. The EEG signal is a fluctuation of voltage between two paired electrodes in respect of time [42] (Figure 3a) and signal amplitude is usually evaluated using peak to peak technique (Figure 3b).

For the evaluation of human emotions, brains response to various stimuli are usually measured and analyzed by five frequency ranges from EEG signals, namely: delta, theta, alpha, beta and gamma. These band waves are omnipresent in different parts of the brain [45,46] and are related to various emotional states (Table 1).

Depending on the object of interest, a variety of different methods can be implemented for the processing and analysis of EEG signals. If the purpose is to evaluate an average level of valence and arousal or to detect the efficiency of applied stimulus, a fast Fourier transformation [48] or latency test can be used [49]. If the purpose is to identify a specific emotion and its strength, statistical methods [50] or machine learning techniques [51] can be implemented. A review of related works based on only EEG signals is provided in Table 2.

From information provided in Table 2 it is seen that main part of researches are focused on the development of more advanced methods for emotion recognition from EEG signals. For this purpose, it is generally not required to provide a real experiment in order to validate the proposed method, because free databases are available with recorded EEG signals under known conditions. One of the most popular databases for EEG signal analysis is DEAP (database for emotion analysis using physiological signals dataset) [62] which contains EEG and other psychological signals from 32 participants stimulated by 40 different one minute length music videos. EEG and peripheral physiological signals recorded using a Biosemi ActiveTwo system. All 32 channels were recorded with sampling frequency 512 Hz. Obtained data relates to results, obtained from self-assessment and from other emotion recognition techniques in order to form reliable dataset, appropriate for use in the future researches. A comparison between the DEAP dataset, MAHNOB-HCI (multimodal database for affect recognition and implicit tagging) [63], EMDB (emotional movie database) [64] and DECAF (multimodal dataset for decoding affective physiological responses) [65] is provided in [66].

Information on the scientific research of emotion recognition using EEG is provided in Table 2. The main focus of activities points to the development of new methods for information extraction from EEG rather than a measurement procedure and therefore opens broad potential for machine learning techniques with IoT capabilities. Moreover, in practice, EEG quite often uses sensor data fusion, which is a basic technique complemented by other sensors and methods. Thus, big data technology with IoT implementation opens new horizons in automated emotions recognition.

### 2.2. Electrocardiography (ECG)

The heart is one of the most critical organs in the human body, and electrocardiography (ECG) is considered to be one of the most powerful diagnostic tools in medicine that is routinely used for the assessment of the functionality of the heart. ECG being a physiological signal is used as the conventional method for noninvasive interpretation of the electrical activity of the heart in real time [67]. Since heart activity is related with human central system ECG is useful not only in analyzing the heart’s activity it can be also used for emotion recognition [68]. 

The ECG recording procedure is described in in detail as follows [69]. The most commonly used technique is known as the 12-lead ECG technique. This technique uses nine sensors placed on the human body (Figure 4a). The positions of the three main sensors are distributed on the left arm (LA), right arm (RA), and on the left leg (LL). The right leg (RL) is connected only by a wire, which should be used as ground for the interconnected sensors. By only having these three sensors, physicians can use a method called 3-lead ECG, which suffers from the lack of information about some parts of the heart but is useful for some emergency cases requiring quick analysis. To obtain a higher resolution, six sensors (V1-V6) are added on the chest (Figure 4a). These sensors also measure to ground (G) on the right leg (RL). Using all the nine sensors and interconnecting them for the 12-lead ECG gives twelve signals, known in biomedical terms as: Lead I, Lead II, Lead III, aVR, aVL, aVF, V1, V2, V3, V4, V5, and V6 (Figure 4b).

The most important points on the ECG signal are the peaks: P, Q, R, S, T, and U [69] (Figure 5b). Each of these peaks is related to the heart activity [69] and it has its own characteristics (Table 3). Emotion recognition using physiological signal is a more complex process compared to EEG because of it’s sensitivity to movement artifacts and the inability to visually perceive emotion from data [70]. In order to eliminate the noises caused by outside factors, such as the movement of the subject during measurement procedure [71], ECG is usually performed in spaces protected from environment effects when the human is in calm state (Figure 5a).

There are main five parameters, as shown in the Table 3, which are often used to evaluate ECG signals. Usually, all five parameters are analyzed only for medical purposes, trying to define abnormal heart activity, and to obtain its deviation parameter. For the recognition of emotions, in most cases, QRS Complex is used, which defines activation of the heart related with human emotional state and is a suitable indicator to recognize main emotions, but there are also difficulties in the emotion recognition due to the fact that this indicator has variant sensitivity to specific emotions. Results of research provided by Cai, Liu, and Hao [73], shows that sadness can be recognized more easily and precisely than emotion of joy. The majority of studies related to ECG based emotion recognition focus on the definition and evaluation of QRS amplitudes and the duration between those waves. Further, there are set of researches focused on the analysis of QT/QTc dispersion [74] which provides proof that this interval is related with anxiety level and can be used as a marker to recognize intense anger.

Main drawback of the 12-lead ECG is that it produces huge amounts of data, especially when used for a long number of hours. Physicians use the 12-lead ECG method because it allows them to view the heart in its three dimensional form, thus enabling the detection of any abnormality that may not be apparent in the 3-lead or 6-lead ECG techniques [69]. ECG application in the automated emotion recognition requires using sophisticated signal processing techniques, which enables detection and extraction of the required parameters from the raw signal. A majority of QRS complex extraction techniques based on assumption that, at the beginning, it is enough to define P or R peaks, and other parameters (Figure 5b) will be estimated using these peaks since the signal shape is stable. There are a huge number of researches available focused on different types of feature extraction methods. Some of those methods include heart rate variability (HRV), empirical mode decomposition (EMD) with-in beat analysis (WIB), FFT analysis, and various methods of wavelet transformations [51]. A detailed overview of various methods used for emotion recognition from ECG is presented in [76]. Analysis of related researches shows the suitability of the ECG technique for precise emotion recognition in the laboratories and predefined stable environments, but fundamental limitations exist that do not allow application of this method for contactless instantaneous emotion recognition. Such methods of emotion evaluation will inevitably be required in future applications in the field of neuromarketing, tutoring, or human–machine interaction. 

Due to the complicity of ECG signal analysis in practical applications, quite often, ECG is used in conjunction with other emotion recognition techniques. A short overview of scientific researches based on ECG is presented in Table 4.

Despite the above-described drawback, ECG remains a powerful and prospective technique for emotion recognitions since it allows to measure signals in the human body which are directly related with emotional states. The fact that many researches focuses on creation of new methods of useful information extraction allows to state that ECG based emotion recognition is a great medium for the implementation of various machine-learning techniques. Machine learning allows for automatically analyzing a huge amount of data and to define relations between measurements performed under various circumstances: states when a human is relaxed or affected by some stimulus. Moreover, due to high precision ECG being complemented by machine learning based signal analysis and processing techniques, it is possible to use for researches of emotion perception mechanisms and for the creation of predictive models based on the long-term monitoring of human behavior and emotional response. 

### 2.3. Galvanic Skin Response (GSR)

The galvanic skin response (GSR), also known as electrodermal activity (EDA) or skin conductance (SC), is a continuous measurement of electrical parameters of human skin. Most often, skin conductions is used as the main parameter in this technique. Electrical parameters of the skin are not under conscious human control [78] since, according to the traditional theory, they depend on the variation of sweat reaction, which reflects changes in the sympathetic nervous system [79]. There is proof that some output signals from sympathetic nervous bursts are followed by the changes of skin conductance [80]. Emotional changes induce sweat reactions, which are mostly noticeable on the surface of the hands fingers and the soles. Sweat reaction causes a variation of the amount of salt in the human skin and this leads to the change of electrical resistance of the skin [81]. When sweat glands becomes more active, they secrete moisture towards the skin surface. That changes the balance of positive and negative ions and affects the electrical currents’ flow property on the skin [82]. 

Skin conductance is mainly related with the level of arousal: if the arousal level is increased, the conductance of the skin also increases. GSR signal amplitude is associated with stress, excitement, engagement, frustration, and anger, and the obtained measurement results correlate with the self-reported evaluation of arousal [83]. Attention-grabbing stimuli and attention-demanding tasks lead to the simultaneous increase of the frequency and magnitude of GSR. So, GSR allows not only to recognize emotions, but also to automatically detect decision making process [84].

In the GSR method, the electrical conductance of the skin is measured using one or two sensor(s) [81] which consist special electrodes containing Ag/AgCl (silver-chloride) contact points with the skin. There is a variety of possibilities for placing electrodes (Figure 6) [85,86], but usually sensors are attached to the fingers, wrist, shoulder, or foot (positions 1, 4, 10, and 6 in Figure 6).

A raw GSR signal contains information about two types of activity: tonic and phasic (Figure 7). The conductivity level of tonic activity changes slowly and individually for each human, and it mainly depends on their skin hydration level, dryness, and autonomic regulation in response to environmental factors such as temperature, for example. Phasic responses are short term peaks in GSR, mostly independent of the tonic level and reflecting reactions of the sympathetic nervous system to emotionally arousing events [87].

Since a GSR signal contains useful information related with its amplitude and frequency, usually, it is analyzed in time and frequency domains by applying various techniques and extracting such statistical parameters as: median, mean, standard deviation, minimum, maximum, as well as ratio of minimum and maximum [78]. The application of traditional signal analysis methods for GSR measurements is complicated by the fact that a signal contains low and high frequency components, and a reaction to the same stimulus is not always identical. Implementing machine learning algorithms, it is possible to increase the precision of emotion recognition and to recognize specific emotions related with the level of arousal, e.g., excitement or stress [81].

Compared to ECG and EEG, GSR gives less information about emotional state, but it has a few important advantages:(i)it requires fewer measuring electrodes, which allows for the easier use of wearable devices and definition of emotional states when a person engages in normal activities;(ii)GSR provides fewer raw data, especially if long term monitoring is performed, this allows to analyse obtained data more quickly and does not require a lot of computational power;(iii)equipment required for GSR measurements is much more simple and cheaper, if special electrodes are available, a measuring device can be assembled using popular and freely available components (ADC converters, microcontrollers, etc.).

The main drawback of the GSR method it is lack of information related to the valence level. This issue is usually solved additionally implementing other emotion recognitions methods, and these complementarily obtained results allows to perform detailed analysis. A short review of researches where GSR is used for emotion recognition is provided in Table 5.

From a review of papers, provided in Table 5, several directions of research are noticeable. The first direction of interest is the development and validation of emotion recognition methods combining GSR and other techniques. The second direction is the development of various wearable sensors. The third direction is the implementation of modern signal processing and analysis techniques in order to create systems, which will be able to define certain emotions with extremely high reliability. An example of such applications is provided in [96], which proposes a stress detection system wherein only two physiological signals are required, namely GSR and heart rate. The study comes up with the conclusions that the best approach combining accuracy and real-time application uses fuzzy logic, modelling the behavior of individuals under different stressing and non-stressing situations, and using that proposed system definitely detected stress by means of fuzzy logic with an accuracy of 99.5%. 

### 2.4. Heart Rate Variability (HRV)

HRV is an emotional state evaluation technique based on the measurement of heart rate variability, which means the beat-to-beat variation in time within a certain period of sinus rhythm (RR interval in Figure 5b). Unlike mean heart rate variance, which is expressed in a period of 60 s, HRV analysis examines the nuance time variance in each cycle of a heartbeat and its regularity [97]. The variability in heart rate is regulated by the synergistic action of the two branches of the autonomous nervous system, namely the sympathetic and parasympathetic nervous system. The heart rate represents the net effect of the parasympathetic nerves, which slow heartrate, and the sympathetic nerves, which accelerate it. These changes are influenced by emotions, stress, and physical exercise [98,99]. Moreover, HRV depends on age and gender, and additional factors include physical and mental stress, smoking, alcohol, coffee, overweight, and blood pressure, as well as glucose level, infectious agents, and depression. Inherited genes also significantly affect heart rate variability. A low HRV indicates a state of relaxation, whereas an increased HRV indicates a potential state of mental stress or frustration [100].

The classical technique for HRV measurements is ECG [97] which measures the primary electro biological signal related with heart activity and provides the ability to define the time between heart pulses by extracting information about the RR interval (Figure 5b) variation in respect to time. Variation of RR interval from ECG signal can be extracted using common peak detection techniques, which allows for defining the duration between each R peak and forms an HRV signal, which expresses the variation of interval between R peaks in respect to time. 

A common method of HVR analysis usually includes analysis methods in time and frequency domains [97]. Various studies based on analyses in one or both domains are shortly reviewed in [101]. The application of HVR for emotions recognition is complicated by the fact that HRV affects other factors, and in order to solve this issue, various signal filtration and feature extraction techniques are implemented. There exist approximately 14 different parameters, which are able to extracted by analyzing HRV. A detailed description of these parameters and their relation with main emotions is presented in [102]. The most common technique used for HRV analyses is the calculation of power spectral density (PSD) of the signal [101]. The PSD represents the spectral power density of a time series as a function of frequency. Typical HRV measurements taken from frequency domain analysis are powers within frequency bands and ratios of powers. The amount of power contained within a frequency band can be obtained by integrating the PSD within the band frequency limits [103].

The main drawback of HRV based on ECG is related to the features of ECG, mainly the complexity of sensors and high requirements for the measurement procedure in order to minimize affects from the environment. An alternative for ECG based HRV is photoplethysmography (PPG). Photoplethysmography is a technique to detect a change of microvascular blood volume in tissues. The principle of this technology is very simple and it requires only a light source and a photodetector. The light source illuminates the tissue and the photodetector measures the small variations in transmitted or reflected light (Figure 8a,b) associated with changes in perfusion in the tissue [99].

The PPG signal (Figure 8b) consists of two main components:(i)the static part of signal depends on the structure of the tissue and the average blood volume arterial and venous blood, and it varies very slowly depending on respiration;(ii)the dynamic part represents changes in the blood volume that occurs between the systolic and diastolic phases of the cardiac cycle [104].

PPG signals, which are analogous voltage values in the time domain, are analyzed using methods similar to those used for the analysis of ECG based HRV. The main difference of the latter to PPG is the filtering of its signal using high-pass filters before defining peaks and forming HRV signal. 

PPG can be performed using only one sensor attached to the finger, or using multiple sensors attached to the right and left ear lobes, index finger pads, and great toe pads [105].

There is a variety of studies proving the successful implementation of this technique and demonstrating its advantages compared to ECG [106,107]. In [105], a comparison between ECG and PPG signal (Figure 9) is presented which proves strict relations between both signals. Delays of PPTp and PPTf in a PPG signal represent the transition time until a pulse from the heart reaches themeasuring point.

Recently, there has been increased interest in remote photoplethysmography (rPPG) whereby it is possible to recover the cardiovascular pulse wave by measuring variations of back-scattered light remotely, using only ambient light and low-cost vision systems [99]. Remote measurements allow to significantly increase human comfort level during the measurement procedure, but this decreases the signal noise ratio and increases the need for more advanced signal processing and analysis algorithms. In [108], machine learning algorithms were implemented in order to the increase precision of HVR measurements performed by smart watches. Results of this research prove that ML is useful tool for PPG measurements data analysis and the extraction of desired features.

Compared to EEG or ECG, HRV (especially based on the PPG technique) is a more comfortable, cheap, and quite universal method. A variety of possible measurement methods in [110] presented a HRV evaluation approach based on heart sound measurements, proving that heart sound correlates well with RR interval from ECG. Another HVR measurement approach using Doppler radar presented by Boris-Lubecke et. al. [111]. In this case, a transceiver transmits a radio wave signal and receives a motion-modulated signal reflected from a human chest which acts as target. Considering that chest movement amplitude in the calm state is about 10 mm due to respiratory and about 0.1 mm due to heart activity, it is possible to extract HRV features from the recorded response signal. A short review of researches focused on emotion recognition using HRV is presented in Table 6.

From Table 6, it is evident that HRV is a quite popular and powerful technique for emotion recognitions. The results of the performed review shows that, in this field, the situation is in contrast to the situation with EEG or ECG, where the attention of researches is directed to the full development of PPG and rPPG techniques, including the development of a novel configuration of wearable PPG sensors, improvement of signal analysis and measurement methods, and the exploration of new application fields.

The main advantage of PPG based HRV consists in the absence of the requirement for special human preparation. Usually, it is enough to touch the active surface of the sensor for a few seconds. The rPPG method provides the possibility of non-contact measurements. Nevertheless, cheap PPG equipment and its accessibility to any potential user is so simple that even the touchscreen of common smartphone can be used as PPG sensor. Mentioned features of this methodology reveal the potential of its implementation in a wide area of applications, especially in the area of human–machine interaction and IoT, since sensors of this type can be easily installed into joysticks and other machine control devices, and can even be hidden for users. 

In special cases, when a multitude of emotions or their detection accuracy has requirements by conditions, the HRV technique needs to be complemented by other techniques, such as ECG, GSR, and data fusion. Such a situation develops the high potential for the applications of big data analysis techniques. 

### 2.5. Respiration Rate Analysis (RR)

Respiratory monitoring data contains useful information about emotional states. Respiration velocity and depth usually vary with human emotion: deep and fast breathing shows excitement that is accompanied by happy, angry, or afraid emotions; shallow and fast breathing shows tension; relaxed people often have deep and slow breathing; shallow and slow breathing shows a calm or negative state. A normal breathing rate in calm states is about 20 times per minute, while in excitement, it can reach up to 40–50 times per minute [118]. The respiration processes is quite complex, and it affects a major part of the body, and due to this many techniques for respiration evaluation exist. Main measurement methods fall into several groups according to measurement principles: manual or semi-automatic breath rate evaluation using simple timers or specialized software applications;methods based on measurements of air humidity fluctuation in exhaled air;methods based on measurements of temperature fluctuation in exhaled air;measurements based on definition of air pressure variation due to respiration;methods based on measurements of variation of carbon dioxide concentration;measurements of variation of oxygen concentration;methods based on measurements of body movements;methods based on measurements of respiratory sounds.

Moreover, it is possible to extract respiratory rate from ECG, PPG, or even blood pressure measurements. All above mentioned methods are explained in detail in [119]. Another very detailed review focused on respiration measurement methods, sensors, and signal processing techniques is provided in [120].

Despite the numerous methods for respiration rate measurement, the popularity of implementation of this technique in the field of emotion recognition is lower in comparison with ECG, GSR, or HRV methods. The main obstacles limiting the application of respiration monitoring are caused by the nature of the signal. Although breath rate depends on emotional state, it can be affected by a variety of external factors, such as human body movement or the level of human fatigue, while environmental conditions, such as air temperature and humidity level also can influence measurement results. Such complex signal requires to implement advanced signal processing techniques, like machine learning algorithms [118] or complement research by using additional measuring methods in order to extract required information from measurement signals. The use of the latter is limited by the fact that a majority of measurement methods requires the use of contact sensors, thus creating discomfort and limitations for normal human activity.

The main advantage of the emotion evaluation technique, based on respiration rate analysis is it’s possibility to implement non-contact measurements methods unlike in EEG or ECG, for example measurements of body movement using video or thermal cameras. In the case of using video camera, signal, which shows respiration rate variation in respect to time, information obtained from tracking displacement of reference point by comparing sequentially, recorded frames by video analysis algorithms. The use of thermal cameras defines respiration rate by analyzing temperature fluctuations near the mouth and nose area caused by exhaled air. 

A short review of researches with respiration rate analysis for emotion recognition is provided in the Table 7. 

The analysis of papers, provided in the Table 7, allows to for the conclusion of emotion recognition based on respiration rate evaluation used as a complimentary method for enforcing other emotion recognition and evaluation methods. Despite the fact that this method is not frequent and carries some functional limitations, it can be successfully applied in cases where the subject takes a fixed position and does not change it significantly during the monitoring period, for example, in the control operation of technological machines or automotive driving. In 2005, Healey and Picard [126] presented methods for collecting and analyzing physiological data during real world driving tasks to determine a driver’s relative stress level. ECG, electromyogram, GSR, and respiration were recorded continuously while drivers followed a set route through open roads. Task design analysis recognized driver stress level with an accuracy of over 97% across multiple drivers and driving days. Provided with a successful example of respiration rate method, let us hope that in the future respiration rate analysis, complemented by other methods can become a big player in emotion analysis within the field of human–machine interaction.

### 2.6. Skin Temperature Measurements (SKT) 

The best bio signal for automatic emotion recognition is signals, which represent a reaction of the autonomic nervous system, which is beyond human control. Skin temperature is one such parameter, related to the human heart activity and sweat reaction. The thermal radiation of a cutaneous surface depends on the perfusion controlled by the autonomic nervous system, which controls the vessels that irrigate the skin. Although the parasympathetic system has an influence through the endothelial cells (in body places like: palmar and plantar surfaces, tip of the nose, sensitive point on the face), the vasomotion is principally regulated by sympathetic noradrenergic fibers, whose activation leads to vasoconstriction and to the decrease of local temperature [127]. In [128], the results prove a good correlation between skin-surface temperature and fingertip blood flow. In [129] it was defined that finger temperature varies due to emotional states and an applied stimulus. Emotions like stress with predominant anxiety, anger, embarrassment, humiliation, joy with anxiety, depression with hostility, guilt, fear of abandonment or fear of conflict over the use of hands for aggressive and sexual purposes, causes decrease of finger temperature. In cases when a patient was not involved in action, but only affected by the speech of another human, experiencing such emotional reactions as anger and anxiety, there was a fall in finger temperature. In addition, a fall of temperature was detected in situations which disturb human safety devices. Similar results were also defined in [130] where it was found that the skin temperature of patients was higher for the expression of low intensity negative emotions compared to the expression of low intensity positive emotions.

In the literature, the most often used temperature measurements methods include: contact method based on the implementation of various semiconductors sensors [117] and non-contact method based on face or full body thermal imaging using infrared cameras [127]. A typical example of skin temperature changes due to an applied stimulus is provided in Figure 10.

Advantage of SKT is possibilities of non-contact measurements, which provides high comfort for the patients and allows eliminating Hawthorne effect (people behave differently, while being observed). Moreover, SKT can be used evaluate emotions not only for humans, but also for animals. In [131,132], stress and body temperature dependencies in animals are discussed, these studies assert that, under stress, animals experience a rise in temperature.

The main drawback of the SKT technique is quite big latency compared to the previously described method. This creates some limitation for this method: a stimulus is required which will take some amount of time and will cause intense emotions, due to this, SKT is well suited to evaluating longer actions like songs or advertising videos, but is not the best choice for evaluating pictures or situations which disappear in a short period of time. The inability to recognize an exact emotion is also a drawback of this method, which can be compensated for by combining SKT with other techniques, but usually it will be less reliable compared with other methods. In [133] presented research, where the input signals were electrocardiogram, skin temperature variation, and GSR, all of which were acquired without much discomfort from the body surface, and can reflect the influence of emotion on the autonomic nervous system. A support vector machine was adopted as a pattern classifier. Correct classification ratios for 50 subjects were 78.4% and 61.8%, for the recognition of three (sadness, anger, stress) and four (sadness, anger, stress, surprise) emotion categories, respectively.

A short summary of researches in which SKT was implemented is provided in Table 8.

From the information provided in Table 8, it is seen that SKT is a popular emotion evaluation method, which fits good with other methods and doesn’t require complicated measurement equipment in the case of contact measurements. This is well suited for the cases where high recognition precision is not required. The results of the performed review points to the focus of researches in this field of the development and evaluation of a new emotion recognition methodology based on SKT measurements, and to the improvement of non-contact emotion evaluation techniques, which are able to define one or a few intense emotions. Such methods have great potential in the future of smart applications and can be useful in medicine, tutoring, human–machine interaction etc.

### 2.7. Electromyogram (EMG)

Electromyography is a technique for evaluating and recording the electrical potential generated by muscle cells [142]. In medicine, this test is used to detect neuromuscular abnormalities, in emotion recognition field it is used to find the correlation between cognitive emotion and physiological reactions [142]. A majority of EMG based researches focus on the analysis of facial expressions due to the hypothesis that facial mimicry contributes to the emotional response to various stimuli. This hypothesis was first announced by Ekman and Friesen in 1978 [143] who described dependencies between simple emotions, facial muscles, and their caused actions (Table 9). Depending on the purpose of analysis, the activity of selected facial muscles (most often: occipitofrontalis, corrugator supercilii, levator labii superioris, zygomaticus major and orbicularis oculi) can be recorded [144].

The EMG procedure is performed by measuring voltages between special electrodes. The EMG is usually done in two steps: in the first step, a baseline is defined (voltage level then human is in calm state) [147]. This level is unique for each person and depends on multiple factors. In the second step, the response to stimulus is measured, and the caused effect evaluated as a ratio between base line and measured value. 

Typical places for electrode location during facial EMG are shown in Figure 11. A huge variety of electrodes can be classified into a few groups according their properties [148]. There are two main types of EMG electrode: surface (or skin electrodes) and inserted electrodes. Inserted electrodes are further classified into two types: needle and fine wire electrodes (Figure 12a,b). Surface electrodes can also be classified into two types: dry electrodes and gelled electrodes (Figure 12c,d).

Needle electrodes are most often used in medical applications. They consist of wire, which is isolated by a special thin tube, and only the end point of the electrode acts as an active contact surface. The advantages of these electrodes include a good signal noise ratio and the possibility to take a precise readout from a relatively small area. Wire electrodes can be made from any small diameter, highly non-oxidizing, stiff wire with insulation. Wire electrodes are extremely fine, they can be implanted more easily, and they are less painful compared to the needle electrodes. 

Gelled surface electrodes contain a gelled electrolytic substance, which allows an electric current from the muscle to pass across the junction between skin, electrolyte, and electrode. Silver chloride (Ag-AgCl) gelled electrodes are used most often. Dry EMG electrodes do not require a gel interface between the skin and the detecting surface. Dry electrodes are usually heavier (>20 g) as compared to gelled electrodes (<1 g), and due to this, special material for fixation of the electrode on the skin is required. The main advantages of surfaces electrodes, compared to needle ones, is that they can be reusable, and they allow non-invasive measurements. Moreover, universal electrodes [150] can be used for EMG, EEG, and ECG procedures just by changing the electrode location and data acquisition device. Drawbacks of the use of surface electrodes is the strict requirements for skin preparation (shaved hair, degreased skin), bigger measurement area, and inefficient signal noise ratio.

Comparing to the latter, procedures of measurement using EMG, EEG, and ECG are similar, but in scientific research, procedures of EMG are seldom used. The main limitation of EMG it is sensitivity to the emotion intensity, however it is a very good technique to detect strong emotions. Nevertheless, small changes of valence and arousal intensity could not be detected, since facial expressions changes only due to strong emotions [151]. The second limitation is the same as for EEG and ECG: This procedure requires the use of contact measurement methods, and therefore it affects comfort level of the persons and creates limitations for their casual activity. In addition, EMG (especially when surface electrodes are used) similarly to ECG, raises requirements for the room in which the procedure is performed: it is necessary to protect from direct sunlight and from electromagnetic noise. Direct sunlight can cause uncontrolled movement of facial muscles; electromagnetic noise can increase noise level in the signal and destroy measurement signals. An advantage of EMG compared to EEG and ECG is the relatively simple analysis of a signal, since various muscles or their groups are affected by different emotions, and separate emotions can be more easily defined analyzing recorded signals.

Comparing emotion recognition and precision EMG gives better results than SKT. In [122] a methodology and a wearable system for the evaluation of the emotional states of car-racing drivers is presented. The proposed approach performs an assessment of the emotional states using facial electromyograms, electrocardiogram, respiration, and GSR. The emotional classes identified are high stress, low stress, disappointment, and euphoria. Support vector machines (SVMs) and an adaptive neuro-fuzzy inference system (ANFIS) have been used for the classification. The overall classification rates achieved by using tenfold cross validation are 79.3% and 76.7% for the SVM and the ANFIS, respectively.

A short summary of researches where the EMG methodology is implemented is provided in Table 10. In a majority of cases, EMG is used in combination with other methods, and researchers have focused on the development of emotion recognition methods and data classification and analysis techniques. 

### 2.8. Electrooculography (EOG)

Electrooculography is a technique for measuring the corneo-retinal standing potential that exists between the front and the back of the human eye. Primary applications appear in ophthalmological diagnosis and in recording eye movements [156]. To measure eye movement, pairs of electrodes are typically placed either above or below the eye, or to the left and right of the eye (Figure 13a). If the eye moves from center position toward one of the two electrodes, an electrical potential appears between those electrodes which corresponds to the eye’s position (Figure 13b) [157]. The idea of implementing EOG for emotion recognition is based on the same hypothesis as EMG, and EOG is often used as a complementary technique. EOG in most cases relies on the detection of eye-blinking and is useful to detect emotions such as stress or surprise [158]. EOG is also useful for assessing fatigue, concentration, and drowsiness [159]. A comparison between response signals from EMG and EOG provided in Figure 14. 

EOG can be applied using contact and non-contact measurement techniques. Contact measurements suitable using EMG electrodes and same equipment. Non-contact measurements can be performed using video camera videooculographysystems (*VOG*) or infrared camera infrared oculography (IROG) [157].

From Figure 14, it is evident that an EOG signal correlates with EMG in time scale, but signal amplitude is much smaller and some latency between vertical and horizontal EOG is noticeable. The complexity of EOG signal processing depends on the measurement method and on the information which can be extracted from the signal. The simplest case is blinking detection, which is represented by peaks in the time-domain signal recorded from electrodes. The detection of exact eye position will require recording the baseline and conscious eye movements in order to have a relation between voltage variation and the position of the eye (Figure 14). The extraction of time-dependent features will require some analysis in the frequency domain, for example, FFT or Wavelet transformation [163]. In the case of non-contact measurement, this method uses analysis under different vision-based object detection and tracking algorithms. 

Compared to EMG, EOG is less powerful technique (it can recognize less amount of simple emotions) but it provides the possibility of non-contact measurements. Disadvantages of EOG and EMG quite similar: if the procedure is performed in a natural office environment, test eye movement can be caused by some unrelated external effect, like bright sunlight, noise, or influence of other persons.

A short summary of researches wherein EOG is implemented as a methodology is provided in Table 11.

The scientific research presented in the Table 11 embraces EOG technology for emotion recognition and their intensity evaluation. A majority of techniques separate positive and negative emotion levels (see [164,165,166]). The evaluation of emotion intensity level remains uncertain for many cases and not comprehensibly described. Video-based systems demonstrate great potential for implementation due to widespread hardware availability as well as the performance of off-line analysis of existing video material.

### 2.9. Facial Expresions (FE) Body Posture (BP) and Gesture Analysis (GA)

In the past decade, there has been a noticeable increase of interest in emotions recognition methods based on the analysis of facial expressions, body posture and gestures. This increase of interest is possibly explained by recent advances in computer vision systems. Emotion recognition methods based on analysis of facial expression, body postures, and gestures are based on the same [143] hypothesis as EMG, claiming that body postures and gestures are also involved in the response of emotions [169,170] and suitable for recognizing the same elementary emotions. A common assumption is that body language is just a different method to express the same basic emotions, e.g., expressed by facial motion. Moreover, the same muscles are used to express emotions in widely different cultures [171]. A summary of main relations between body postures and emotions is provided in Table 12. 

Advantages and difficulties of these methods stem from the fact that, in the human body, there are plenty of reference points which should be monitored. One of the most advanced commercially available system from Imotion, namely the Facial Action Coding System (FACS) [172], uses various combinations of 64 parameters for emotion recognition. Measurements of facial expressions, body posture, and gestures are usually performed using computer vision systems and analysis algorithms, which can track movements of selected reference points. Such a measurement technique also has preferences in the emotion recognition field, since it allows for the performance of non-contact measurements and produces quite reliable results. In [173,174] presented research cases where emotion recognition accuracy in a random scenario was 60–86%.

Since the above presented method is based on the same hypothesis as EMG, similar limitations exist: (i)it recognizes only strong emotions which last some amount of time, response to weak emotions or to very short not intense stimulus does not create the noticeable facial movements or change in body posture;(ii)the possibility exists that changes in human motion or facial expressions are due to environmental effects.

Also, there exist a few drawbacks related to the measurement methods: (i)huge amount of data is created while tracking a lot of reference points;(ii)track of body posture: it is difficult to define the exact position of a reference point, which is covered by clothes, and in this case, special marks for vision systems should be implemented.

Despite the mentioned drawback, facial expressions, body posture, and gesture tracking remain promising techniques in the emotion recognition field, especially taking into account recent advances in computer vision systems, big data analysis, and machine learning techniques. A short summary of researches involving the analysis of facial expressions, body posture, and implemented gestures is provided in Table 13. 

From Table 13, it is evident that, in a majority of researches, facial expressions, body posture, and gestures analysis methods were used together, and were even complemented by other techniques in order to improve recognition accuracy. Also in the literature, there are presented cases where these methods were used together with not so common techniques: for example, speech analysis [175,177,178]. 

Comparing facial expressions, body posture, and gestures analysis methods with previously described methods, it can be stated that these methods are the most promising in future applications. Especially in practical cases, which do not require extremely high accuracy and sensitivity due to their wide applicability, a large number of measurable parameters, as well as advances in video analysis and large data processing capabilities, allows for the implementation of a multimodal approach. 

One of the most promising implementations of facial expression analysis is the Internet of Things. IoT objects, which respond to users’ emotional states, can be used to create more personalized user experiences. The IoT covers fields as diverse as medicine, advertising, robotics, virtual reality, diagnostic software, driverless cars, pervasive computing, affective toys, gaming, education, working conditions and safety, automotive industry, home appliances, etc., which will significantly benefit from emotion-sensing technology.

The analysis of facial expression, body movements, and gestures represent a contactless method applicable for mass and individual emotion recognition. All techniques are simple to collect data but require sophisticated video frame analysis in dynamics and sculpted surface analysis for static frame content. Methods for facial recognition and gesture recognition are typically separate, but material for them is the same. Methods are available in real time systems and in the off-line mode, and therefore in-depth analysis, the development of reactions in the time progression, and human test progression easily possible. As video-based systems, these techniques demonstrate the potential to grow in the future. 

All positive features of mentioned methods are limited by a great amount of data, which raises the requirement for data storing, processing, and cross-analysis time. Cloud computing and IoT represent some solutions, but then mobile data transmission can be a bottleneck for the recent situation. Another drawback would be low accuracy in the intensity level definition, as temperament is a key parameter in gesture and facial expression.

## 3. Signal Analysis and Features Extraction Methods

Reliability, precision, and speed of emotion evaluation strongly depend not only on the used measurement method and sensor, but also on the applied signal processing and analysis technique. In this chapter, we will provide a review of the most commonly used signal analysis and feature extraction methods (Table 14).

In [184,185,186,187,188,189,190,191,192] analyzed a number of various emotions using different measurement methods and feature extraction techniques. For example, in [184] introduced the overall paradigm for their multimodal system that aims at recognizing its users’ emotions and responding to them accordingly depending upon the current context or application. They described the design of the emotion elicitation experiment they conducted by collecting, via wearable computers, physiological signals from the autonomic nervous system (GSR, HRV, SKT) and mapping them to certain emotions (sadness, anger, fear, surprise, frustration, and amusement). They showed the results of three different supervised learning algorithms that categorize collected signals in terms of emotions, and generalize their learning to recognize emotions from new collections of signals. Overall, three algorithms, namely the k-nearest neighbor (KNN), discriminant function analysis (DFA), and Marquardt backpropagation algorithm (MBP), could categorize emotions with 72.3%, 75.0%, and 84.1% accuracy, respectively.

Li and Chen [185] proposed to recognize emotion using physiological signals obtained from multiple subjects from the body surface. Four physiological signals, namely ECG, SKT, GSR, and respiration rate, were selected to extract features for recognition. Canonical correlation analysis was adopted as a pattern classifier, and the correct classification ratio is 85.3%. The classification rates for fear, neutral, and joy were 76%, 94%, and 84% respectively. 

A new approach to enhance driving safety via multi-media technologies by recognizing and adapting to drivers’ emotions (neutrality, panic/fear, frustration/anger, boredom/sleepiness) with multi-modal intelligent car interfaces is presented in [192]. A controlled experiment was designed and conducted in a virtual reality environment in order to collect physiological data signals (GSR, HRV, and SKT) from participants who experienced driving-related emotions and states (neutrality, panic/fear, frustration/anger, and boredom/sleepiness). KNN, MBP, and resilient backpropagation (RBP) algorithms were implemented to analyze the collected data signals and to find unique physiological patterns of emotions. RBP was the best classifier of these three emotions with 82.6% accuracy, followed by MBP with 73.26%, and KNN with 65.33%.

In [186] presented an artificial intelligence based system which could detect the early onset of fatigue in drivers using HRV as the human physiological measure. The detection performance of the neural network was tested using a set of ECG data recorded under laboratory conditions. The neural network gave an accuracy of 90%. 

In [187] presented classification of three emotions (boredom, pain, and surprise) by using four machine learning algorithms (linear discriminate analysis (LDA), classification and regression tree (CART), self-organizing map (SOM), and support vector machine (SVM)). GSR, ECG, HRV, and SKT as physiological signals were acquired for one minute before emotional state as a baseline and for 1–1.5 min during emotional states. For emotion classification, the difference values of each feature-subtracting baseline from the emotional state were used for machine learning algorithms. The result showed that an accuracy of emotion classification by SVM was the highest. In the analysis of LDA, the accuracy of all emotions was 78.6%, and in each emotion, boredom was recognized by LDA with 77.3%, pain 80.0%, and surprise 78.6%. CART provided accuracy of 93.3% when it classified all emotions. In boredom, accuracy of 94.3% was achieved with CART, 95.9% in pain, and 90.1% in surprise accuracy rate of LDA was 78.6%, 93.3% in CART, and SOMs provided accuracy of 70.4%. The result of emotion classification using SOM showed that, according to orders of boredom, pain and surprise recognition accuracy of 80.1%, 65.1%, and 66.2% were obtained by SOM correspondingly. Finally, the result of emotion classification using SVM showed an accuracy rate of 100.0%.

User-independent emotion recognition method with the goal of recovering affective tags for videos using electroencephalogram, pupillary response and gaze distance presented in [188]. Initially, 20 video clips were selected with extrinsic emotional content from movies and online resources. Ground truth was defined based on the median arousal and valence scores given to clips in a preliminary study using an online questionnaire. Based on the participants’ responses, three classes for each dimension were defined. The arousal classes were calm, medium aroused, and activated and the valence classes were unpleasant, neutral, and pleasant. The best classification accuracies of 68.5% for three labels of valence and 76.4% for three labels of arousal were obtained using a modality fusion strategy and a SVM.

In [189] presented a specific emotion induction experiment to collect five physiological signals of subjects including ECG, GSR, blood volume pulse, and pulse. The support vector regression (SVR) method was used to train the trend curves of three emotions (sadness, fear, and pleasure). Experimental results show that the proposed method achieves a recognition rate up to 89.2%.

In [190] proposed and investigated a methodology to determine the emotional aspects attributed to a set of computer aided design (CAD) tasks by analyzing the CAD operators’ psycho-physiological signals. Psycho-physiological signals of EEG, GSR, and ECG were recorded along with a log of CAD system user interactions. A fuzzy logic model was established to map the psycho-physiological signals to a set of key emotions, namely frustration, satisfaction, engagement, and challenge and the results were analyzed. The correlations between fuzzy model outputs and reported emotions are 84.18% (frustration), 76.83% (satisfaction), 97% (engagement), and 97.99% (challenge) respectively.

A novel approach for the multimodal fusion of information from a large number of channels to classify and predict emotions is presented in [191]. The multimodal physiological signals are 32 channels EEG, eight-channel GSR, blood volume pressure, and respiration pattern, SKT, EMG, and EOG. The experiments are performed to classify different emotions (terrible, love, hate, sentimental, lovely, happy, fun, shock, cheerful, depressing, exciting, melancholy, mellow) from four classifiers. The average accuracies are 81.45%, 74.37%, 57.74%, and 75.94% for SVM, MLP, KNN, and MMC classifiers respectively. The best accuracy is for ‘depressing’ with 85.46% using SVM.

An evaluation of the accuracy of provided methods, presented in Table 14, outlined three technologies with the best accuracy of emotion recognition, namely ECG, EEG, and GSR. A noticeable positive influence implementing FUZZY logic for the recognition of emotion intensity level is supported by respectable research [190]. An increase in the number of recognized emotions sharply decrease the quality and reliability of recognition, and this proposes a new roadmap for emotion recognition process planning. Three emotions recognition accuracy can reach even 100%, but their intensity level is still not defined uniquely. Research confirmed better recognition of negative rather than positive emotions in a majority of methods.

## 4. Discussion

The selection of measurement methods and sensors is a complex process in which a huge set of questions are presented. There are multiple choices for physiological parameters to measure as well as physical principles of obtaining signals. Technology of measurements in relation to particular sensors creates a huge set of possibilities to select. Multiple attempts to classify emotions, sensors, and universal selection algorithms have been made. Some attempts are presented in [193] and we try to fill this gap with the current proposal. In contrast to physical measurements, obtaining emotions from human body parameter measurements creates a tough problem. Therefore, basic sensor selection methods become unclear due to a lack of classification methods and functional relations between sensors and desired emotions.

We conclude by providing the classification of emotion recognition measurement methods (Figure 15), which allows to fulfil the selection procedure proposed in [193] as a two-step procedure. The first step consists of the selection of measurement parameters and methods, while the second step realizes the selection of sensors.

We assume that, in the beginning of method selection, it is necessary to define if we are interested in a conscious or unconscious response, or maybe in both methods simultaneously. Research based on conscious responses are relatively simple and does not require any special hardware, but it requires a lot of attention to prepare questionnaires. In contrast, results from self-evaluation are not so reliable, and there exists a possibility that a person will not recognize their own emotions correctly or will provide imprecise answers to uncomfortable questions. Methods based on unconscious responses usually provide more reliable results, but they require multiple attempts for measuring procedures and raise high requirements for hardware. 

Methods based on unconscious responses provide many choices and we propose the selection of electrical or non-electrical parameter measurement. As all reactions in the human body are controlled by electric signals generated in the central nervous system, we can state that electric parameters are primary entities, which gives mostly precise results, and the measurement of non-electrical signals give reactions of the human body affected by electric signals. On the other hand, electrical signals can be measured using only contact measurement methods, and of course there is the possibility to send a signal to an acquisition unit using wireless techniques, but despite this, there remain some limitations for human activity during measurement procedures.

Measurements of electrical parameters have two features: it is possible to use methods based on direct (self-generating) sensors, when is measured signal crated by central nervous system (EEG, ECG, HRV, EMG, EOG), or measurements based on modulating sensors when changes in human body modulates properties of the sensor (GSR). From theory, it is known that direct sensors are more precise [193] but they can a little affect the signal (especially signal with small amplitude) since they take part of the power form it. On the other hand, modulating sensors will have some latency, which depends on properties of individual sensor. 

Measurement methods based on measurements of non-electrical parameters usually suffer from lack of accuracy and latency but their main advantage is possibility to perform non-contact methods without limiting human activity and they better fits for field applications and for approximate emotional state evaluation. 

Recent researches in the field of emotion recognitions shows that there is no method that is ideal for one case and the best solution is multimodal analysis, as presented in [194] or in [195] using several methods they complements each other and allows to achieve a higher reliability of obtained results. 

A noticeable methodological problem in all emotion recognition techniques is lack of united conception of dataset. Researchers choose control group sizes, compositions, experimental time, and periods arbitrarily or based on possibility. Features in each emotion recognition methodology are different, but certain description for reliable standards, covering dataset issues begs for definition. This would release unnecessary resources, used by research with reliable results on the output. 

Signal processing and analysis techniques also play important roles in the selection of methods and sensors. In a majority of cases, the effectiveness of emotion recognition depends on the applied procedures of signal processing and analysis. For example, from ECG data, it is possible to extract information about HRV and respiration rate. Recent research shows that the most powerful techniques applied for emotion recognition are multi-criteria analysis based on statistical methods (ANOVA) or on machine learning algorithms. 

Summing up, we can state that the interest in emotion recognition and practical implementation of this technique is steadily increasing and finds more areas of application. Detailed research available in public sources is mostly focused on the physiological side of this object. We found a lack of research and unified classification focused on the engineering part of this question, for example, missing rationale related to measurement methods, measurement uncertainties, and clear specification as to which method, sensor, processing, and analysis techniques are best suited for recognizing a particular emotion.

The future of such analysis can present a background for future systems with emotion recognition. The sensors and methods for human emotion recognition along with computer vision, speech recognition, deep learning, and related technologies have demonstrated tremendous progress in the IoT field. Due to this, the understanding of human emotions has also experienced distinct progress [29]. 

The study and development of systems and devices constitutes affective computing. This is a means for recognizing, interpreting, processing, and simulating the influences on people [196]. Machinery for recognizing, expressing, modelling, communicating, and responding to information about emotions as well as certain affective computing instances have been built globally by numerous researchers [197]. Innovative understanding of the self and better, advanced human communications have become possible by the advances of affective computing technology. This promises new technologies for reducing stress, rather than increasing it. Management requires measurement, as is popularly said. The real-time skills provided by computers are complex and challenging. These skills allow them greater understanding and intelligent responses to human emotions, which are complicated but occur naturally and, for the same reason, are expressed naturally. The range for their use covers a number of fields such as those in the fields of human sciences like neuroscience, physiology, and psychology [198]. The state of the art for multi-modal affect analysis frameworks, however, lacks a comprehensive discussions in surveys of available scholarly literature [199]. 

The realization of mood sensor technology is expected on an annual basis. Efforts in research and development will increasingly aim at contactless technology involved in measuring emotions, despite the on-body devices and/or voice/facial recognition software currently required by most existing human emotion recognition sensors, methods, and technologies. However, regardless of these efforts, there is very slow movement towards the humanization of the IoT with human emotion recognition methods and/or sensors at present [29]. Consequently, this research represents an attempt at introducing the idea of humanizing the IoT and affective computing systems by applying human emotion recognition sensors and methods to academic and business communities. The confirmation of such is by the IoT and affective computing systems, which were developed by the authors of this research [25,26,27,28].

## 5. Conclusions and Future Trends

Emotion recognitions is a powerful and very useful technique for the evaluation of human emotional states and predicting their behavior in order to provide the most suitable advertising material in the field of marketing or education. In addition, emotion recognition and evaluation is very useful in the development process of various human machine interaction systems.

Relations between particular emotions and human body reactions have long been known, but there remain many uncertainties in selecting measurement and data analysis methods. There are eight methods most used in that field, which are based on measurements of various parameters and an innumerable majority of data analysis methods and attempts for practical applications. 

In this review, we observed more than 160 scientific articles and provided the classification of AEE methods, using a summarized description of common emotion recognition methods and various attempts to improve the reliability of its results. This paper also provides an engineering view to AEE methods and their reliability, sensibility, and stability.

In the near future, a combination of those methods and implementation of machine learning for data analysis seems to be an extremely powerful combination, which will create breakthroughs in practical application in all fields starting from advertising and marketing and finishing with industrial engineering applications.

## Figures and Tables

**Figure 1 sensors-20-00592-f001:**
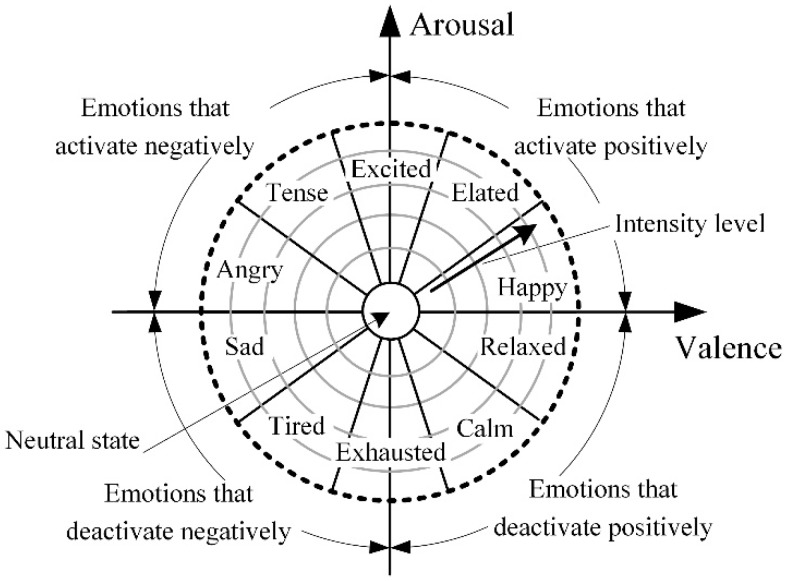
Russel’s circumplex model of emotions.

**Figure 2 sensors-20-00592-f002:**
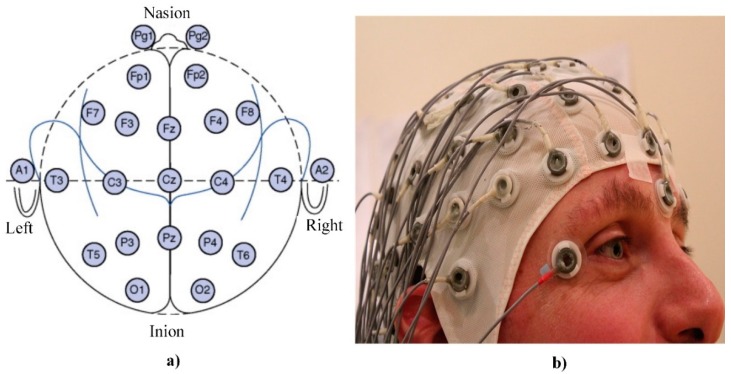
Electroencephalography (EEG) measurements: (**a**) distribution of EEG electrodes on human scalp [39]; (**b**) special headset with installed electrodes [40].

**Figure 3 sensors-20-00592-f003:**
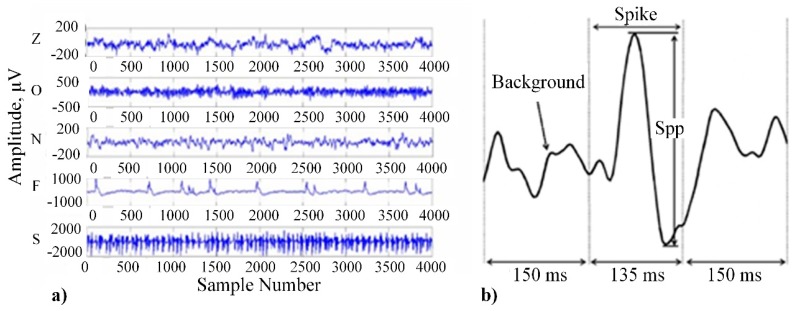
EEG signal: (**a**) example of raw data [43]; (**b**) peak to peak signal amplitude evaluation technique [44].

**Figure 4 sensors-20-00592-f004:**
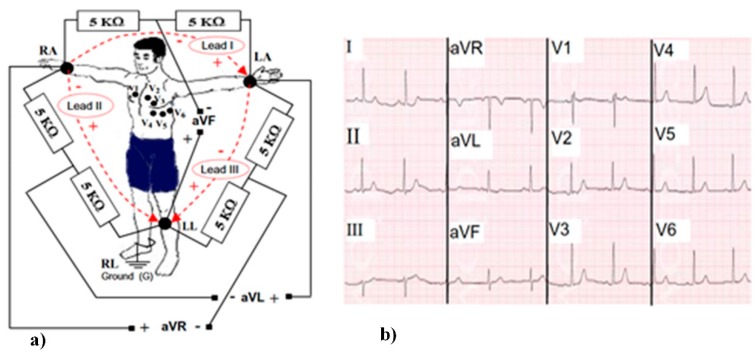
Schematic representation of electrocardiography (ECG) [69]: (**a**) 12-lead ECG: RA, LA, LL, RL; (**b**) example of ECG signals.

**Figure 5 sensors-20-00592-f005:**
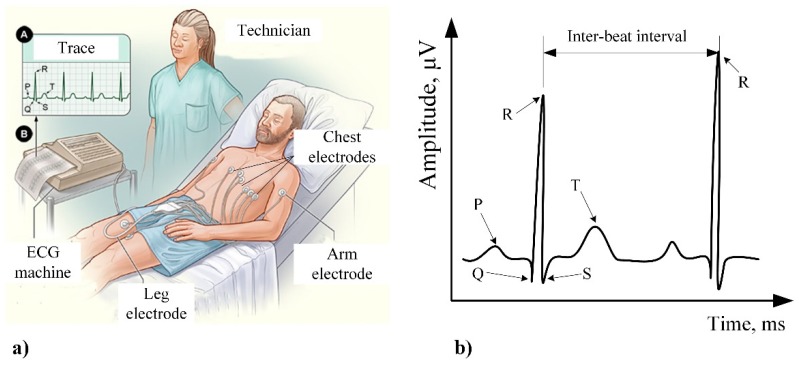
ECG procedure [72]: (**a**) typical set up; (**b**) Main parameters of an ECG heartbeat signal.

**Figure 6 sensors-20-00592-f006:**
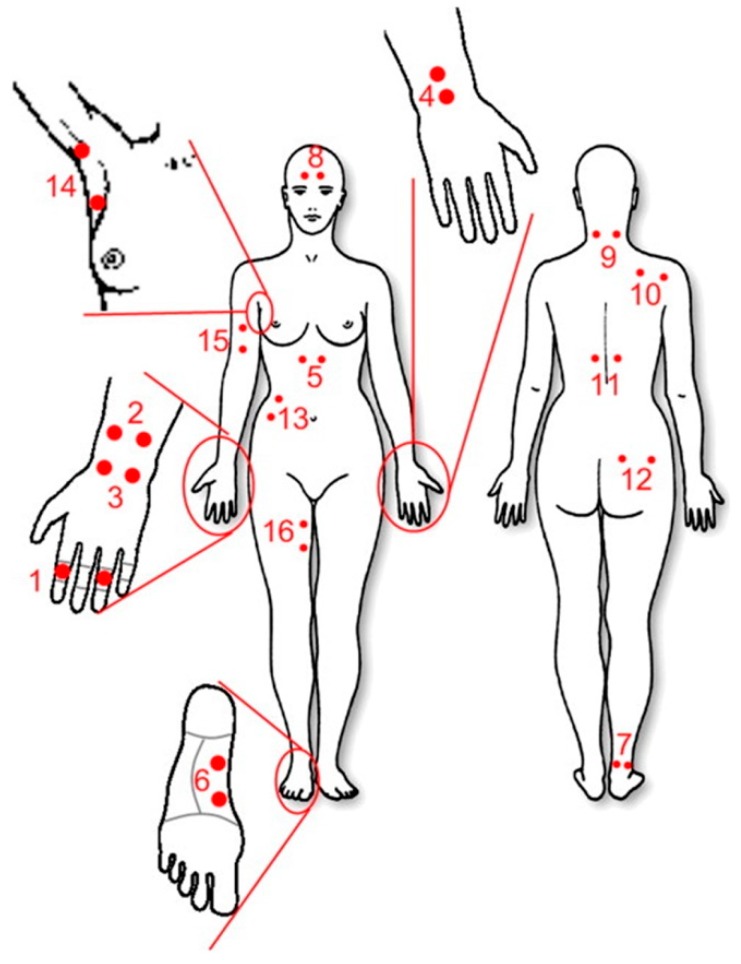
Possible places for attaching GSR electrodes [86].

**Figure 7 sensors-20-00592-f007:**
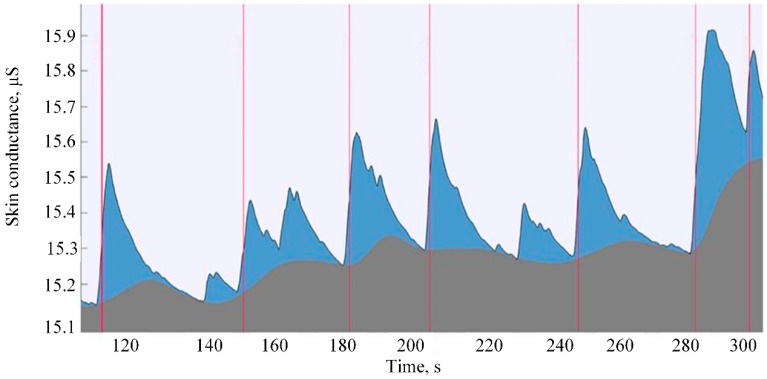
Example of raw GSR signal. The blue area indicates the phasic component of the signal; grey area represents the tonic component. The red line indicates the trigger (moment of delivery of the stimulus) [88].

**Figure 8 sensors-20-00592-f008:**
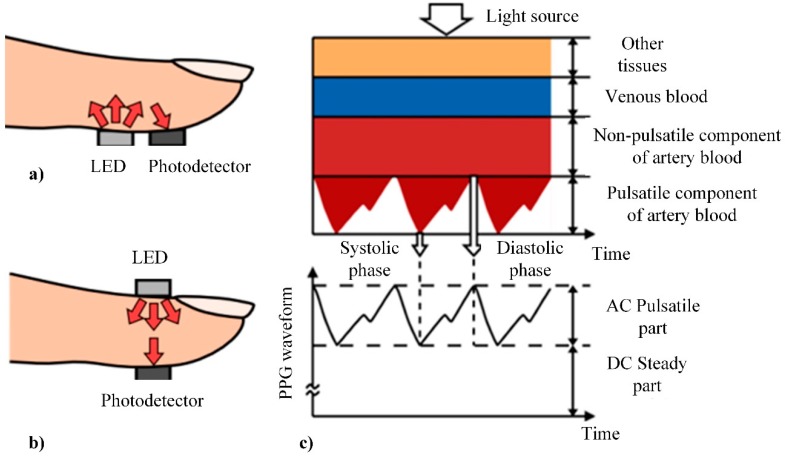
Principle of photoplethysmography (PPG) [104]: (**a**) reflective mode; (**b**) transmitting mode; (**c**) example of PPG signal.

**Figure 9 sensors-20-00592-f009:**
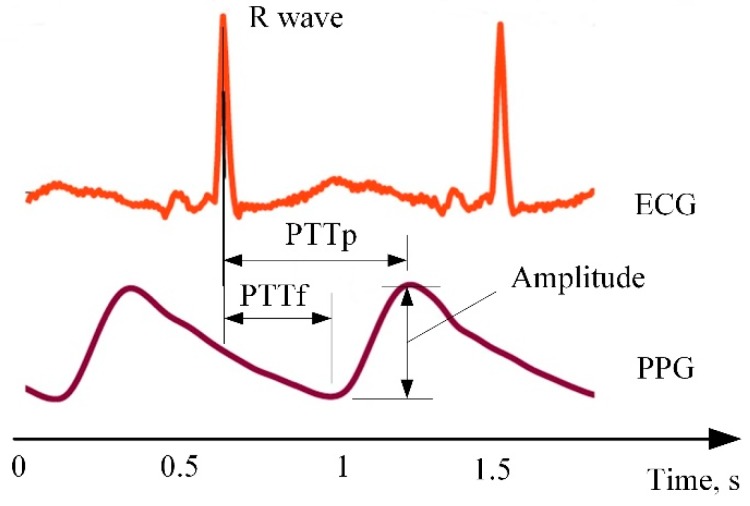
Comparison between ECG and PPG signals [109].

**Figure 10 sensors-20-00592-f010:**
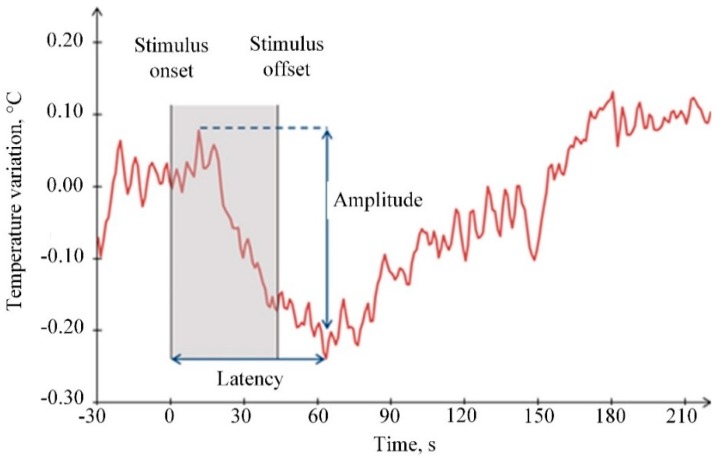
Example of skin temperature change due to applied stimulus [127].

**Figure 11 sensors-20-00592-f011:**
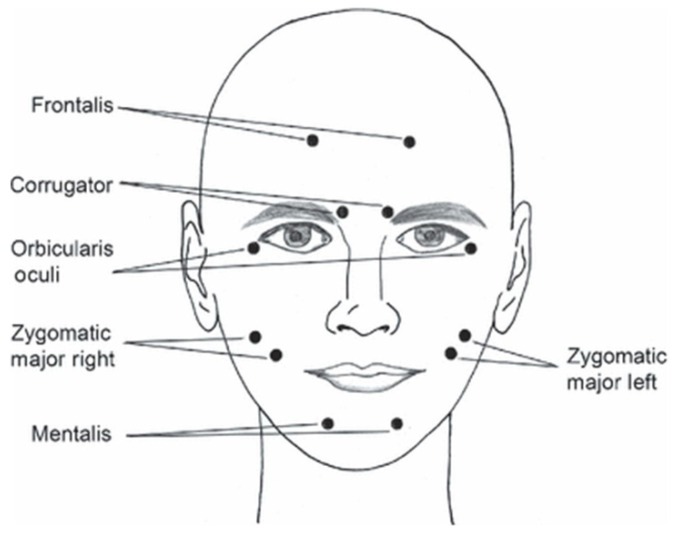
Facial electromyography [149]: location of electrodes.

**Figure 12 sensors-20-00592-f012:**
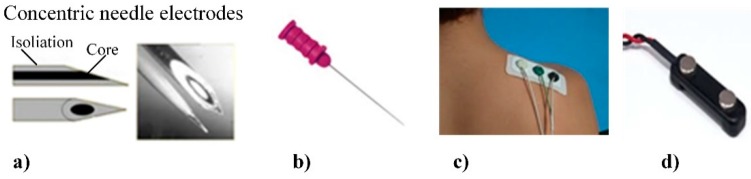
Example of EMG electrodes [148]: (**a**) needle electrode; (**b**) fine wire electrode; (**c**) gelled electrodes; (**d**) dry electrodes.

**Figure 13 sensors-20-00592-f013:**
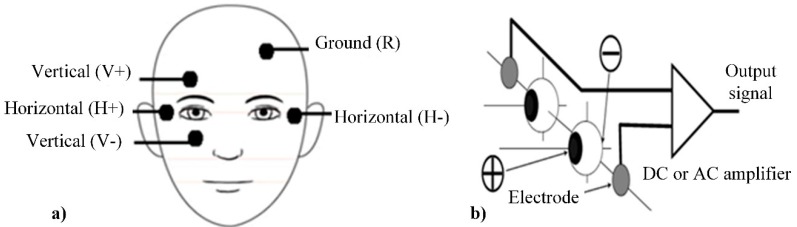
Principle of electrooculography (EOG): (**a**) electrode placement scheme [160]; (**b**) measurement principle [161].

**Figure 14 sensors-20-00592-f014:**
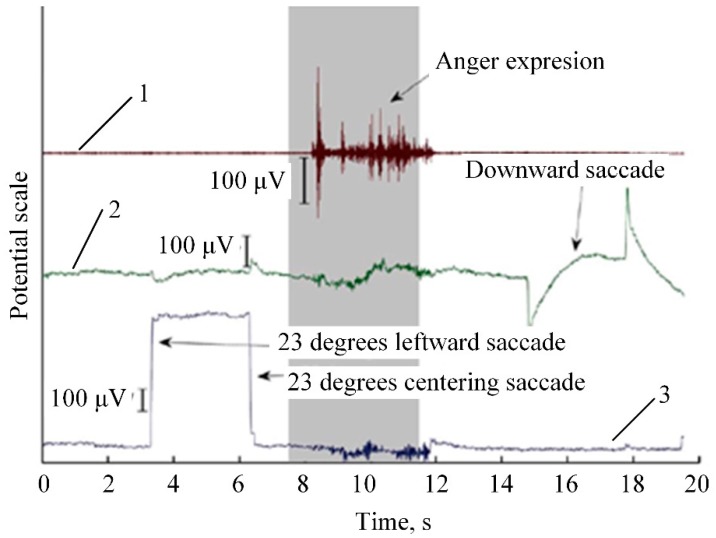
Comparison between EOG and EMG signals during three different, sequential actions [162]: 1—Corrugator supercilii EMG; 2—vertical EOG; 3—horizontal EOG.

**Figure 15 sensors-20-00592-f015:**
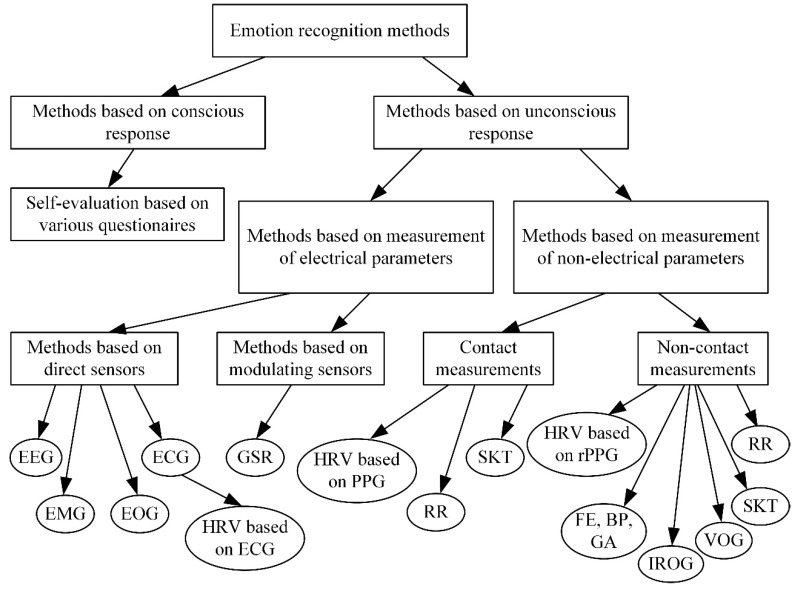
Classification of measurement methods for emotions recognition.

**Table 1 sensors-20-00592-t001:** Classification of brain waves [47,48].

Type of Waves	Related Emotional State	Short Description
Delta (*δ*) (0.5–4 Hz)	Strong sense of empathy and intuition	The slowest brain waves often associated with sleep. Multiple frequencies in this range are accompanied by the release of human growth hormone, which is useful in healing. These waves produced in the waking state show an opportunity to access the subconscious activity.
Theta (*θ*) (4–8 Hz)	Deep relaxation, meditation	Mainly adults produce the theta brain waves, when the person is in the light sleep or in dreams. These waves normally appear with closing the eyes and disappears with opening of eyes. Frequency of these waves is mainly associated stress relief and memory recollection. Twilight conditions can be used to reach deeper meditation resulting in improved health, as well as increasing creativity and learning capabilities
Alpha (*α*) (8–16 Hz)	Creativity, Relaxation	These waves mostly present during the state of awake relaxation with eyes closed. Alpha is the resting state for the brain. Activity of alpha waves decreases in response to all types of motor activities. Alpha waves aid overall mental coordination, calmness, alertness, mind/body integration, and learning
Beta (*β*) (16–32 Hz)	Beware, Concentration.	The beta waves are produced when the person is in an alert or anxious state, and it is a dominant rhythm. Usually, they are generated in the frontal and central part of the brain. In this state, brains can easily perform: analysis, preparations of the information, generate solutions and new ideas.
Gamma (*γ*) (32 Hz-above)	Regional Learning, Memory and Language Processing and Ideation.	These waves are emitted when a person is in the abnormal condition or there will be some mental disorder. Gamma brainwaves are the fastest of brain waves and relate to simultaneous processing of information from different brain areas. Numerous theories have proposed that gamma contributes directly to brain function, but others argue that gamma is better viewed as a simple byproduct of network activity

**Table 2 sensors-20-00592-t002:** Review of scientific researches focused on emotions recognition and evaluation using only electroencephalography (EEG) signals.

Aim	Emotions	Hardware and Software	Ref.
Creation of emotion classification system using EEG signals.	High/low arousal and valence.	5 channels *wireless* headset *Emotiv Insight*	[52]
Creation of new emotions evaluation technique based on a three-layer EEG-ER scheme.	High/low arousal and valence	Electro-cap (Qucik-Cap 64) from NeuroScan system (Compumedics Inc., Charlotte, NC, USA)	[53]
Research of Relief-based channel selection methods for EEG-based emotion recognition	Joy, fear, sadness, relaxation	−	[54]
Creation of an intelligent emotion recognition system for the improvement of special students learning process	Happy, calmness, sadness, scare	Emotiv-EPOC System. 14 electrodes with two reference channels were used	[55]
Automated human emotions recognition from EEG signal using higher order statistics methods.	High/low arousal and valence	The EEG input signals were provided by the DEAP database	[56]
Creation of new methodic for recognition of human emotions	High/low arousal and valence	Multi-channel EEG device was used	[57]
New EEG-based emotion recognition approach with a novel time-frequency feature extraction technique is presented	High/low arousal and valence	The EEG signals provided by the DEAP dataset	[58]
New deep learning framework based on a multiband feature matrix (MFM) and a capsule network (CapsNet) is proposed.	High/low arousal, valence and dominance	The DEAP dataset was used	[59]
New cross-subject emotion recognition model based on the newly designed multiple transferable recursive feature elimination are developed	High/low arousal, valence and dominance	32 channel data from DEAP dataset was used to validate the proposed method	[60]
Presented novel approach based on the multiscale information analysis (MIA) of EEG signals for distinguishing emotional.	High/low arousal and valence	The EEG input signals were provided by the DEAP dataset	[61]

**Table 3 sensors-20-00592-t003:** Description of main parameters of electrocardiography (ECG) signal [75].

Parameter	Duration, s	Amplitude, mV	Short Description
P	~0.04	~0.1–0.25	This wave is a result from strial contraction (or depolarization). P wave that exceeds typical values might indicate atria hypertrophy.
PR	0.12–0.20	–	The PR interval measured from the start of the P wave to the start of Q wave. It represents the duration of atria depolarization (contraction).
QRS Complex	0.08–0.12		The QRS complex measured from the start of Q wave to the end of S wave. It represents the duration of ventricle depolarization (contraction). If the duration is longer, it might indicate the presence of bundle branch blocks.
QT/QTc	~0.41		It is measured from the start of the Q wave to the end of T wave. QT interval represents the duration of contraction and relaxation of the ventricles. Duration of QT/QTc varies inversely with the heart rate.

**Table 4 sensors-20-00592-t004:** Review of scientific researches focused on emotion recognition and evaluation using ECG.

Aim	Emotions	Methods	Hardware and Software	Ref.
Study focuses on emotion recognition for service robots in the living space	High/neutral/low valence. Negative arousal categorized into: sadness, anger, disgust, and fear	**ECG**	Wireless bio sensor RF-ECG	[1]
This research suggests an ensemble learning approach for developing a machine learning model that can recognize four major human emotions	Anger; sadness; joy; and pleasure	**ECG**	Spiker-Shield Heart and Brain sensor	[51]
Creation of new methodology for the evaluation of interactive entertainment technologies.	Level of arousal	**ECG**, galvanic skin response (GSR), electromyography of the face, heart rate	Digital camera, ProComp Infiniti system and sensors, BioGraph Software from Thought Technologies.	[4]
Presentation of new AfC methodology capable of recognizing the emotional state of a subject.	High/low valence and arousal	**ECG**, EEG	B-Alert X10 sensor (Advanced Brain Monitoring, Inc., USA)	[77]
Proposed new method for the automatic location of P-QRS-T wave, and automatic feature extraction	Joy and sadness	**ECG**	BIOPAC System MP150	[73]

**Table 5 sensors-20-00592-t005:** Review of scientific researches focused on emotions recognition and evaluation using GSR.

Aim	Emotions	Methods	Hardware and Software	Ref.
Stress level evaluation in computer human interaction.	Stress	**GSR**, eye activity	Mindfield eSense sensor, Tobii eye-tracker environment (Tobii Studio)	[34]
Creation of textile wearable system, which is able to perform an exosomatic EDA measurement using AC and DC methods.	Level of arousal	**GSR**	Textile electrodes, from Smartex s.r.l. (Pisa, Italy), installed into special glove	[89]
Research of proposed methodologies for emotions recognition from physiological signals	Valence and arousal levels	**GSR**, heart rate	Polar-based system, Armband from Bodymedia	[90]
Assessment of human emotions using peripheral as well as EEG physiological signals on short-time periods	High/neutral/low valence and arousal	**GSR**, EEG, blood pressure	Biosemi Active II system (http://www.biosemi.com), plethysmograph to measure blood pressure	[91]
Assessment of human emotion from physiological signals by means of pattern recognition and classification techniques	High/low valence and arousal	**GSR**, EEG, blood pressure, a respiration, temperature	Biosemi Active II device (http://www.biosemi.com), GSR sensor, plethysmograph, respiration belt and a temperature sensor	[92]
Creation of wearable system for measuring emotion-related physiological parameters	–	**GSR**, heart rate, skin temperature	Originally designed glove with installed sensors	[93]
Validation of new method for the emotional experience evaluation extracting semantic information from the autonomic nervous system	High/low valence and arousal	**GSR**, ECG, heart rate,	Bodymedia Armband, InnerView Research Software 4.1 from Bodymedia	[94]
Development of two state emotion recognition engine for mobile phone	Pleasant unpleasant	**GSR**, Photoplethysmogram (PPG), Skin temperature	–	[95]

**Table 6 sensors-20-00592-t006:** Review of scientific researches focused on emotions recognition and evaluation using HRV.

Aim	Emotions	Methods	Hardware and Software	Ref.
Objective of this study was to recognize emotions using EEG and peripheral signals.	High/low valence and arousal	**HRV**, EEG, GSR, blood pressure, respiration	Biosemi Active II system (http//www.biosemi.com). GSR sensor, plethysmograph, respiration belt	[112]
Creation of new method for the identification of happiness and sadness	Happiness and sadness	**HRV**, skin Temperature (SKT),	SKT sensor, PPG sensor	[113]
Aim of this project was to design a noninvasive system which will be capable of recognizing human emotions using smart sensors	Happiness (excitement), sadness, relaxed (neutral), and angry	**HRV**, skin temperature SKT, GSR	Custom made PPG sensor, DS600 temperature sensor by Maxim—Dallas semiconductor, Custom made GSR sensor	[114]
This article describes the development of a wearable sensor platform to monitor a mental stress.	Mental stress	**HRV**, GSR, respiration	Heart rate monitor (HRM) (Polar WearLink+; Polar Electro Inc.), Respiration sensor (SA9311M; Thought Technology Ltd.), GSR sensor (E243; In Vivo Metric Systems Corp.). EMG module (TDE205; Bio-Medical Instruments, Inc.)	[115]
This paper investigated the ability of PPG to recognize emotion	High/low valence and arousal	**HRV**	PPG sensor	[116]
The present research proposes a novel emotion recognition framework for the computer prediction of human emotions using wearable biosensors	Happiness/Joy, anger, fear, disgust, sadness	**HRV**, GSR, SKT, Activity recognition	PPG sensor, GSR sensor, SKT, fingertip temperature; EMG gyroscopes and accelerometer for activity recognition, Android smartphone for data collection	[117]

**Table 7 sensors-20-00592-t007:** Review of scientific researches focused on emotions recognition and evaluation using respiration rate measurements.

Aim	Emotions	Methods	Hardware and Software	Ref.
This paper investigates computational emotion recognition using multimodal physiological signals	Positive, negative and neutral arousal	PPG, GSR, **respiration rate** skin temperature	Pulse oximeter (PP-CO12, TEAC Co.) GSR, (PPS-EDA, TEAC Co, AP-U030, TEAC Co.), respiration rate sensor (AP-C021, TEAC Co.), temperature sensor clip (AP-C050, TEAC Co.)	[121]
This paper introduces an automated approach in emotion recognition, based on several bio signals	Stress, disappointment, euphoria	Electromyograms (EMGs), ECG, **respiration rate**, and GSR.	EMG textile fireproof sensors; the ECG and respiration sensors on the thorax; the GSR textile and fireproof sensor placed special glove	[122]
To compare time, frequency, and time-frequency features derived from thermal infrared data discriminates between self-reported affective states of an individual in response to visual stimuli drawn from the international affective pictures system	High/neutral/low valence and arousal	Facial thermal infrared data, blood volume pulse (BVP), and **respiratory effort**	FLIR Systems ThermaCAM (SC640) long wavelength infrared (LWIR) camera, piezo crystal respiratory effort sensor belt 1370G by Grass Technologies, BVP sensor (PPS) by Grass Technologies Atmospheric temperature sensor HS-2000D	[123]
Design experimental stand which is used in monitoring human-system interaction	High/low arousal	GSR, Electromyography (EMGs,) **respiration rate**, EEG, blood-volume pulse, temperature	SC-Flex/Pro sensor, MyoScan Pro EMG, Respiration rate sensor, EEG-Z sensor, HRV/BVP Flex/Pro sensor Temperature sensor	[124]
This paper aims at assessing human emotion recognition by means of the analysis of HRV with varying spectral bands based on respiratory frequency	High/neutral/low arousal	ECG, **respiration rate**, blood pressure (BP), skin temperature (ST) GSR	EEG, blood pressure, skin temperature and GSR sensors	[125]

**Table 8 sensors-20-00592-t008:** Review of scientific researches focused on emotions recognition and evaluation using SKT.

Aim	Emotions	Methods	Hardware and Software	Ref.
Present App for smartphones *CaptureMyEmotion*, which can improve learning process of autistic children.	High/low arousal	**SKT**, GSR, motion analysis	Q sensor from Affectiva (www.affectiva.com).	[134]
Proposed a new method for evaluating fear, based on nonintrusive measurements obtained using multiple sensors	Fear	EEG, **SKT**, eye blinking rate	EEG device (Emotiv EPOC), commercial thermal camera (ICI 7320 Pro) commercial web-camera (C600) and a high-speed camera	[135]
Study infant emotion rely on the assessment of expressive behavior and physiological response	Joyful emotion	**SKT**	Thermal imaging system (TH3104MR, NEC, Sanei)	[136]
To demonstrate that the effects of particular emotional stimuli depend not only on physical temperatures but also on homeostasis/thermoregulation.	Emotionally warm or emotionally cold state	-	–	[137]
Present new methodology which offers a sensitive and robust tool to automatically capture facial physiological changes	High/low valence and arousal	**SKT**, ECG, GSR	ECG and GSR National Instruments (NI) devices, infrared camera FLIR A615	[138]
Evaluation of possibility of wireless determination of skin temperature using iButtons	–	–	iButton (type DS1921H; Maxim/Dallas Semiconductor Corp., USA)	[139]
Present a new approach how to analyze the physiological signals associated with emotions	Sadness, amusement, fear, anger, surprise	**SKT**, GSR	BodyMedia, SenseWear armband	[140]
Present a new StressCam methodology for the non-contact evaluation of stress level	Stress	**SKT**	Infrared camera	[141]

**Table 9 sensors-20-00592-t009:** Relations between emotions and facial expressions [145,146].

Emotion	Involved Muscles	Actions
Happiness	Orbicularis oculi, Zygomaticus major	Closing eyelids, pulling mouth corners upward and laterally
Surprise	Frontalis, Levator palpebrae superioris	Raising eyebrows, raising upper eyelid
Fear	Frontalis, Corrugator supercilii, Levator palpebrae superioris	Raising eyebrows, lowering eyebrows, raising upper eyelid
Anger	Corrugator supercilii, Levator palpebrae superioris, Orbicularis oculi	Lowering eyebrows, raising upper eyelid, closing eyelids
Sadness	Frontalis, Corrugator supercilii, Depressor angulioris	Raising eyebrows, lowering eyebrows, depressing lip corners
Disgust	Levator labii superioris, Levator labii superioris alaeque nasi	Raising upper lip, raising upper lip and wrinkling nasal skin

**Table 10 sensors-20-00592-t010:** Review of scientific researches focused on emotions recognition and evaluation using EMG.

Aim	Emotions	Methods	Hardware and Software	Ref.
Research of possibility to reliably recognize emotional state by relying on noninvasive low-cost EEG, EMG, and GSR sensors	High/low valence and arousal	EEG, GSR, **EMG**, HRV	BrainLink headset, Neuroview acquisition software, Shimmer GSR+Unit Shimmer EMG device, a plethysmograph	[152]
Present a new approach for monitoring and detecting the emotional state in elderly	High/low arousal	EDA, HRV, **EMG**, SKT, activity tracker	EDA-custom made sensor, a plethysmograph, SKT resistance temperature detector, 3-axis accelerometer	[153]
Present a model that allows to determine emotion in real time	High/low valence and arousal	**EMG**, GSR	ProComp Infiniti Bio-signal Encoder, GSR sensor	[154]
Present a methodology and a wearable system for the evaluation of the emotional states of car-racing drivers	Anger, fear, disgust, sadness, enjoyment and surprise	**EMG**, GSR, ECG, respiration rate	EMG textile fireproof sensors; ECG and respiration sensors on the thorax of the driver; the GSR sensor in the glow	[122]
Present fully implemented emotion recognition system including data analysis and classification	Joy, anger, pleasure, sadness	**EMG**, ECG, GSR, respiration rate	Four-channel EMG, ECG, GSR, respiration rate bio sensor	[155]

**Table 11 sensors-20-00592-t011:** Review of scientific researches focused on emotions recognition and evaluation using EOG.

Aim	Emotions	Methods	Hardware and Software	Ref.
Present a novel strategy (ASFM) for emotions recognitions	Positive, neutral, negative emotions	EMG, **EOG**	Off-line experiment was performed using SEED datasets	[164]
Present a novel approach for a sensor-based E-Healthcare system,	Positive, neutral, negative emotions	**EOG**, IROG	Neuroscan system (Compumedics Neuroscan, Charlotte, NC, USA), infrared camera with the resolution of 1280 × 720	[165]
Proposed a new approach towards to the recognition of emotions using stimulated EOG signals	Positive, neutral, negative emotions	**EOG**	Customized EOG data acquisition device, Ag/AgCl electrodes	[166]
The proposed system introduces an emotion recognition system, based on human eye movement	Happy, sad, angry, afraid, pleasant	**EOG**	Video-based eye trackers	[167]
Present a novel strategy of eye movement analysis as a new modality for recognizing human activity.	Arousal level	**EOG**	Commercial system Mobi from Twente Medical Systems International (TMSI)	[168]

**Table 12 sensors-20-00592-t012:** Relations between emotions and body posture [175,176].

Emotions	Gestures and Postures
Happiness	Body extended, shoulders up, arms lifted up or away from the body
Interest	Lateral hand and arm movement and arm stretched out frontal
Surprise	Right/left hand going to the head, two hands covering the cheeks self-touch two hands covering the mouth head shaking body shift–backing
Boredom	Raising the chin (moving the head backward), collapsed body posture, and head bent sideways, covering the face with two hands
Disgust	Shoulders forward, head downward and upper body collapsed, and arms crossed in front of the chest, hands close to the body
Hot anger	Lifting the shoulder, opening and closing hand, arms stretched out frontal, pointing, and shoulders squared

**Table 13 sensors-20-00592-t013:** Review of scientific researches focused on emotions recognition and evaluation using analysis of facial expressions, body posture and gestures.

Aim	Emotions	Methods	Hardware and Software	Ref.
Presentation of ASCERTAIN-a multimodal database for implicit personality and Affect recognition using commercial physiological sensors.	High/low valence and arousal	GSR, EEG, ECG, HRV, **facial expressions**	GSR sensor, ECG sensor, EEG sensor, webcam to record facial activity Lucid Scribe software	[179]
Creation of personalized tool for a child to learn and discuss her feelings	Real time arousal and stress level	**Facial expression recognition**	Smartphone camera, application CaptureMyEmotion	[180]
This paper aims to explore the limitations of the automatic affect recognition applied in the usability context as well as to propose a set of criteria to select input channels for affect recognition.	Valence and arousal, interest, slight confusion, joy, sense of control	GSR, **facial expressions**,	Infiniti Physiology Suite software; standard internet camera and video capture software from Logitech, Noldus FaceReader, Morae GSR recorder	[181]
This study proposes a new method that involves analysis of multiple data considering the symmetrical characteristics of face and facial feature points	Fear	Movement of **facial feature** points such as eyes, nose, and mouth	FLIR Tau2 640 thermal cameras, NIR filter, Logitech C600 web-camera	[182]
Present a novel method, for computerized emotion perception based on posture to determine the emotional state of the user.	Happiness, interest, boredom, disgust, hot anger	**Body postures**	C++ in Ubuntu 14.04. Kinect for Microsoft Xbox 360 and OpenNI SDK	[176]
To propose a novel method to recognize seven basic emotional states utilizing body movement	Happiness, sadness, surprise, fear, anger, disgust and neutral state	**Gestures and body movements**	Kinect v2 sensor	[183]

**Table 14 sensors-20-00592-t014:** Analysis of previous studies on emotion recognition.

Emotions	Measurement Methods	Data Analysis Methods	Accuracy	Ref.
Sadness, anger, stress, surprise	ECG, SKT, GSR	SVM	Correct-classification ratios were 78.4% and 61.8%, for the recognition of three and four categories, respectively	[133]
Sadness, anger, fear, surprise, frustration, and amusement	GSR, HRV, SKT	KNN, DFA, MBP	KNN, DFA, and MBP, could categorize emotions with 72.3%, 75.0%, and 84.1% accuracy, respectively	[184]
Three levels of driver stress	ECG, EOG, GSR and respiration	Fisher projection matrix and a linear discriminant	Three levels of driver stress with an accuracy of over 97%	[126]
Fear, neutral, joy	ECG, SKT, GSR, respiration	Canonical correlation analysis	Correct-classification ratio is 85.3%. The classification rates for fear, neutral, joy were 76%, 94%, 84% respectively	[185]
The emotional classes identified are high stress, low stress, disappointment, and euphoria	Facial EOG, ECG, GSR, respiration,	SVM and adaptive neuro-fuzzy inference system (ANFIS)	The overall classification rates achieved by using tenfold cross validation are 79.3% and 76.7% for the SVM and the ANFIS, respectively.	[122]
Fatigue caused by driving for extended hours	HRV	Neural network	The neural network gave an accuracy of 90%	[186]
Boredom, pain, surprise	GSR, ECG, HRV, SKT	Machine learning algorithms: linear discriminate analysis (LDA), classification and regression tree (CART), self-organizing map (SOM), and SVM	Accuracy rate of LDA was 78.6%, 93.3% in CART, and SOMs provided accuracy of 70.4%. Finally, the result of emotion classification using SVM showed accuracy rate of 100.0%.	[187]
The arousal classes were calm, medium aroused, and activated and the valence classes were unpleasant, neutral, and pleasant	ECG, pupillary response, gaze distance	Support vector machine	The best classification accuracies of 68.5 percent for three labels of valence and 76.4 percent for three labels of arousal	[188]
Sadness, fear, pleasure	ECG, GSR, blood volume pulse, pulse.	Support vector regression	Recognition rate up to 89.2%	[189]
Frustration, satisfaction, engagement, challenge	EEG, GSR, ECG	Fuzzy logic	84.18% for frustration, 76.83% for satisfaction, 97% for engagement, 97.99% for challenge	[190]
Terrible, love, hate, sentimental, lovely, happy, fun, shock, cheerful, depressing, exciting, melancholy, mellow	EEG, GSR, blood volume pressure, respiration pattern, SKT, EMG, EOG	Support Vector Machine, Multilayer Perceptron (MLP), K-Nearest Neighbor (KNN) and Meta-multiclass (MMC),	The average accuracies are 81.45%, 74.37%, 57.74% and 75.94% for SVM, MLP, KNN and MMC classifiers respectively. The best accuracy is for ‘Depressing’ with 85.46% using SVM. Accuracy of 85% with 13 emotions	[191]

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
