# Peer review of "Human Emotion Recognition: Review of Sensors and Methods"

_sensors, 2020, doi:10.3390/s20030592_

Round 1

Reviewer 1 Report

This review is quite interesting, extensive and well organized. The English needs to be revised by a native speaker, there are really lots of typos. I was surprised that the authors didn't include the health domain as a possible domain of application of emotion recognition techniques. Studies in Psychology and Medicine are obviously concerned by automatic recognition of emotions, as patients may show abnormalities in emotional reactivity: examples include older adults with apathy, people with brain damage to areas of the brain involved in emotional processing, people with personality disorders, and so on. I would suggest adding some references to the vast domain of health in the introduction and discussion.

Author Response

Please, review improved manuscript.

Reviewer 2 Report

Paper can be accepted after following corrections:

- Please explain the details in figure 2b.

- Fine wire electrode idea should be explained in more details. Figure 14 b is misleading.

Author Response

New improved version uploaded

Reviewer 3 Report

This article summarized the human emotion recognition methods of using different kinds of sensors, and authors observed more than 160 scientific articles for discussing AEE methods.

This article seems like a summary report, which only focuses on collecting the related work without any further discussing or experiments, and almost all of the figures and tables are cited from other published papers, no novel methods, no unique opinion and view, no theoretical analysis and verification process. This article contains so many unpersuasive references, and some of these reference papers are outdated and lack of sufficient validation. In particular, background and main ideas are supposed to be supported by evidence of academic authority references. In fact, if authors want to write a review article, a more comprehensive analysis and experimental verification are very necessary and not just a simple list of theories. Meanwhile, there is no algorithm or model detailed description all over the article and lack of technical process analysis. Moreover, these is no clear and complete theoretical system from the article structure, the authors just list relevant technologies from 2.1 to 2.8, respectively. And section 3 is a short discussion of the section 2’s methods combination without any further potential relevance analysis, pros and cons discussion and personal views. Finally, section 4 is the conclusion, without experiment section. Actually, the article’s lack of experimental analysis is the biggest problem. According to the further research of potential relevance discussion in section 3, the section 2’s methods can be evaluated based on different data processing mode, emotional evaluation, action principle and other factors. I mean, there is lack of in-depth analysis of these methods.

Author Response

This article seems like a summary report, which only focuses on collecting the related work without any further discussing or experiments, and almost all of the figures and tables are cited from other published papers, no novel methods, no unique opinion and view, no theoretical analysis and verification process.

Answer: We included our unique opinion and expanded discussions sections by describing possible implementation of automated human emotion recognition in the field of Internet of things, which is quite novel

This article contains so many unpersuasive references, and some of these reference papers are outdated and lack of sufficient validation. In particular, background and main ideas are supposed to be supported by evidence of academic authority references.

Answer: We have slightly revised the list of literature including publications from the more known and valuable sources. A large amount of used references can be explained by the fact that aim of this paper is not only review the well-known methods and applications but also includes a cases of implementing unique equipment, measurement methodology, or data analysis methods, which from our point of view may become valuable in the future, when in this field more often will be used machine learning and others modern IT technologies.

In fact, if authors want to write a review article, a more comprehensive analysis and experimental verification are very necessary and not just a simple list of theories. Meanwhile, there is no algorithm or model detailed description all over the article and lack of technical process analysis.

Answer: Our article focuses mainly on review of existing sensor and methods more than on deep analysis of information provided by other research. Our article is more like collection of application cases and expected results using one or another method. Such format should be acceptable for researches which works in this field and for them are necessary to find appropriate method for emotion recognition. Deep analysis and experimental evaluation of each method are performed by other researches to which articles we providing links in the summing up tables in the end of each subchapter (2.1-2.8).

Moreover, these is no clear and complete theoretical system from the article structure, the authors just list relevant technologies from 2.1 to 2.8, respectively. And section 3 is a short discussion of the section 2’s methods combination without any further potential relevance analysis, pros and cons discussion and personal views.

Answer: We provide analysis vertically – along sensor technologies and horizontally – over emotion parameters. Sensitivity, stability and implementation conditions of provided methods are analyzed in the paper, so structure points to researcher, which looks for a method, suitable for his purpose or check on suitability of method to the aim of researcher. We providing not only list of relevant technologies, we also overview of researches related to the uses of each individual method, which allows for the potential read to form opinion about applications possibilities of each method, on same time we providing direct link to more specified material if appears some doubts about technical issues or etc. In the end of each subchapter there are provided a short description of main advantages and disadvantages of each method.

Section 3 which provides not only discussions but also our proposed classification of emotion recognition methods, which should be useful selecting methods and sensors for emotion recognition. Moreover, we improved this section by including review of trends of possible future implementations.

Finally, section 4 is the conclusion, without experiment section. Actually, the article’s lack of experimental analysis is the biggest problem.

Answer: Since it is review article about sensors and methods no experimental research is included into content. But there provided references to the researches about each mentioned cases. Extremely detailed or experimental analysis of mentioned cases goes beyond the scope of this review article.  

Section 4 are included in paper according to instructions provided by publisher where it is declared that conclusion ‘’section is not mandatory, but can be added to the manuscript if the discussion is unusually long or complex‘‘.

According to the further research of potential relevance discussion in section 3, the section 2’s methods can be evaluated based on different data processing mode, emotional evaluation, action principle and other factors. I mean, there is lack of in-depth analysis of these methods.

Answer: Our focus in this paper is directed from the researcher’s point of view rather from analytical point of view to every particular method. In other hand, very deep analysis of all mentioned sensors is so vast; that it fit only in the entire book, otherwise that would make focus point unreadable. Conception of this paper is to provide systematization of used methods, sensors and their usability rather than deep analysis of each methodology.

Round 2

Reviewer 3 Report

Emotional analysis for IoT is really a useful and very broad application prospect in the future, especially for the robotics, marketing, education and entertainment industries. However, from 2.1 to 2.8, some of these sensors are not suit for IoT, and some types of sensors have already widely used because of the low price such as ECG, HRV, FE, BP, GA, but review table 3, 6, 7 without any signal result analysis to emotional expression except basic description. That means, in addition to the use of sensors, we pay more attention to the performance of the effect. In fact, the bottleneck problems in this area right now are that how to transform the sensor signals into knowledge, and what signal analysis results are universally accepted. But, in this paper, there is no discussion of methods (except in the section 2.1’s last paragraph, ML, data fusion, big data technology technique list), and there is no clear conclusion on some results of sensor signal analysis. More important is that there is no evaluation standard of datasets, the number of people tested, emotional triggering mode, testing environment and conditions, effectiveness, etc. Maybe this is a good summary report of the basic theory, but we want to know more valuable content in a review article.

Author Response

Dear Reviewer,

thank you for your valuable review. Your efforts in reading and raising some remarks really increase level of understanding of prepared paper.

We have answered you remarks point by point, where changed places in the paper marked in cyan color. Number of changed and added text is significant, therefore better is to read it directly in the improved uploaded paper.

With respect in the name of authors,

Vytautas Bucinskas